# An adenosine derivative prevents the alterations observed in metabolic syndrome in a rat model induced by a rich high-fat diet and sucrose supplementation

Enrique Chávez[1☯], Alejandro Rusbel Aparicio-Cadena[1☯], Gabriela Velasco-Loyden[1], María Guadalupe Lozano-Rosas[1], Mariana Domínguez-López[1], Amairani Cancino-Bello[1], Nimbe Torres[2], Armando R. Tovar[2], Alejandro Cabrera-Aguilar[1], Victoria Chagoya-de Sánchez[1]*

1 Instituto de Fisiología Celular, Departamento de Biología Celular y del Desarrollo, Universidad Nacional Autónoma de México, Mexico City, Mexico, 2 Departamento de Fisiología de la Nutrición, Instituto Nacional de Ciencias Médicas y Nutrición "Salvador Zubirán", Mexico City, Mexico

☯ These authors contributed equally to this work.
* vchagoya@ifc.unam.mx

## Abstract

Metabolic syndrome is a multifactorial disease with high prevalence worldwide. It is related to cardiovascular disease, diabetes, and obesity. Approximately 80% of patients with metabolic syndrome have some degree of fatty liver disease. An adenosine derivative (IFC-305) has been shown to exert protective effects in models of liver damage as well as on elements involved in central metabolism; therefore, here, we evaluated the effect of IFC-305 in an experimental model of metabolic syndrome in rats induced by a high-fat diet and 10% sucrose in drinking water for 18 weeks. We also determined changes in fatty acid uptake in the Huh-7 cell line. In the experimental model, increases in body mass, serum triglycerides and proinflammatory cytokines were induced in rats, and the adenosine derivative significantly prevented these changes. Interestingly, IFC-305 prevented alterations in glucose and insulin tolerance, enabling the regulation of glucose levels in the same way as in the control group. Histologically, the alterations, including mitochondrial morphological changes, observed in response to the high-fat diet were prevented by administration of the adenosine derivative. This compound exerted protective effects against metabolic syndrome, likely due to its action in metabolic regulation, such as in the regulation of glucose blood levels and hepatocyte fatty acid uptake.

## Introduction

Metabolic syndrome is a multifactorial pathology that can include obesity, dyslipidemia, glucose intolerance, insulin resistance and arterial hypertension and can be associated with type 2 diabetes mellitus, metabolic dysfunction–associated steatotic liver disease (MASLD), and cardiovascular disease [1, 2].

**Data Availability Statement:** All relevant data are within the paper and its Supporting Information files.

**Funding:** ARAC is recipient of a Consejo Nacional de Humanidades, Ciencias y Tecnologias (CONAHCYT)-Mexico fellowship 720321. VCdS received financial support from Programa de Apoyo a Proyectos de Investigacion e Innovacion Tecnological-Universidad Nacional Autonoma de Mexico (PAPIIT-UNAM) IN214419. The funders had no role in study design, data collection and analysis, decision to publish, or preparation of the manuscript.

**Competing interests:** The authors have declared that no competing interests exist.

Metabolic syndrome rates are estimated from those associated with obesity and related diseases. According to the World Obesity Atlas 2022, in 2030, the prevalence of obesity will be one billion people [3]. Overweight and obesity are observed primarily in industrialized or developing countries, such as Mexico [4].

The primary stimulus of metabolic syndrome is an increase in body weight, although insulin resistance associated with an inflammatory response promoted by cytokines and other mediators from adipose tissue and liver is also considered one of the principal causes [5–7]. Moreover, disequilibrium of lipid, glucose, insulin, and proinflammatory cytokine concentrations have been related to the overproduction of reactive oxygen species and mitochondrial dysfunction [7, 8]. There is an increased risk of developing MASLD in patients who have metabolic syndrome [9]. Damage to hepatocytes begins with excessive capture of free fatty acids; furthermore, the synthesis of lipids from carbohydrates is increased because the enzymes involved in these processes are directly regulated by glucose and insulin [10].

Insulin resistance has been reversed using antagonists of adenosine 2b receptor (A2BR), which also diminishes glucose levels in diabetic mice [11]. In addition, A2BR activation induces IL-6 production in macrophages and endothelial cells in diabetic rats [12]. Furthermore, adenosine administration alleviates hyperlipidemia and hyperinsulinemia, increases insulin sensitivity and downregulates the mRNA expression of the A2B receptor [13–15].

IFC-305 or adenosine aspartate is a derivative of adenosine that has been shown to have the same hepatoprotective effects as adenosine (200 mg/kg) at a dose 4 times lower (50 mg/kg) in addition to prolonging the half-life of adenosine. The hepatic maximum concentration of adenosine is at 15 min after its intraperitoneal administration (200 mg/kg), and at 30 min, the concentration is approximately 0.5 μmol/g of tissue, being almost undetectable at 120 min. In contrast, the IFC-305 maximum concentration in the liver tissue (approximately 2.7 μmol/g of tissue) is found 30 min after its administration (50 mg/kg), and at 120 min, the concentration is maintained at 1.7 μmol/g of tissue. This effect may be one of the reasons why a lower dose of IFC-305 than adenosine is needed to obtain the same hepatoprotective effect [16]. Adenosine has been shown to be hepatoprotective by preventing and reversing liver damage in different experimental models [17, 18]. This compound restores hepatic function, including by inducing liver regeneration in a model of experimental cirrhosis; prevents and reverses hepatocellular carcinoma induced by diethylnitrosamine; restores mitochondrial function; induces autophagy [19–22].

According to the abovementioned findings, the aim of this work was to evaluate the effects of IFC-305 on the alterations observed in metabolic syndrome in rats induced by a high-fat diet and 10% sucrose in drinking water for 18 weeks. The high-fat diet and sucrose increased body mass, serum triglycerides and proinflammatory cytokines compared to those in control rats. However, the adenosine derivative significantly prevented these changes, including the histological alterations, and inhibited the alterations in glucose and insulin tolerance, enabling the rats to regulate their glucose levels. The *in vitro* uptake of fatty acids was also prevented by IFC-305 treatment.

## Materials and methods

### Chemicals

IFC-305 (adenosine aspartate) was prepared with adenosine free base (MP Biomedicals, LLC, Illkirch, France) and l-aspartic acid (MP Biomedicals, Inc., Eschwege, Germany) as previously described in patent MX 207422 [23].

**Table 1. HF diet composition.**

| Ingredients | (g/kg) |
|---|---|
| Casein | 240 |
| L-cystine | 3 |
| Corn starch | 239.03 |
| Maltodextrin | 102.67 |
| Sucrose | 77.783 |
| Soy oil | 70 |
| Cellulose | 50 |
| Mineral Mix | 35 |
| Vitamin Mix | 10 |
| Choline | 2.5 |
| Lard | 170 |
| TBHQ | 0.013 |

The components were first mixed; then, lard and oil were added to obtain a homogeneous mass. It was stored at 4°C until use.

## Animals

Male *Wistar* rats weighing 180–220 g (7 to 8 weeks old) were obtained from and housed at the Animal Facility of the Universidad Nacional Autónoma de México (UNAM). The animals were housed in pairs per cage under controlled conditions (22 ± 2°C, 50–60% relative humidity, and 12 h light-dark cycles). The animals were used according to institutional guidelines, and the protocol was approved by the Comité Institucional para el Cuidado y Uso de Animales de Laboratorio del Instituto de Fisiología Celular (VCHH53-14) and the Mexican Official Norm (NOM-062-ZOO-1999). Four experimental groups were created (n = 6) as follows: the control group, the high-fat diet + 10% sucrose in drinking water (HFS) group, the HFS + IFC-305 (HFS+IFC-305) group and the IFC-305 group. The control and IFC-305 groups were fed a standard diet (AIN-93) and were administered saline solution i.p. and IFC-305 (50 mg/kg, daily i.p.), respectively. The HFS and HFS+IFC-305 groups were fed a modified standard diet supplemented with fat from an animal source (HF diet, Table 1) plus sucrose solution 10% *ad libitum* and were administered saline solution and IFC-305 (50 mg/kg, daily i.p.), respectively. All groups were treated for 18 weeks. The AIN-93 diet components and modifications were established as previously described [24, 25]. The animals were euthanized using sodium pentobarbital (Pisa, Mexico), and the livers were removed and frozen in liquid nitrogen for further analysis. The sections were fixed in 4% paraformaldehyde for histological analysis.

## Electron microscopy

Liver samples for electron microscopy were fixed in glutaraldehyde (6%) and stained with osmium tetroxide (1% phosphate-buffered saline solution) as previously reported [21].

## Growth curves

Body mass was measured twice a week during treatment for analysis of weight gain.

## Glucose and insulin tolerance tests

At the end of the treatment, blood glucose levels were measured. Blood glucose concentrations in tail vein samples were measured using a glucometer (Accu-Chek). Beforehand, the animals were fasted overnight.

Glucose and insulin tolerance tests were performed by gavage administration of glucose solution (3 g/kg) in 3 mL of water and i.p. injection of rapid-acting insulin (100 IU/ml, 0.2 IU/kg), respectively. First, the glucose concentration in the blood was measured at 0, 15, 30, 60, 120 and 180 minutes after glucose solution administration. The insulin tolerance curve was determined using the same fasting conditions.

## Serum levels of triglycerides, LDL cholesterol, and insulin

Blood samples were collected by cardiac puncture. Serum samples were obtained by centrifugation at 10,000 rpm for 15 min at 4°C. Triglycerides, LDL cholesterol, and insulin levels were determined using commercial kits (Spinreact™ refs: 1001310, 41023, EZRMI-13K, respectively). The instructions provided by the manufacturers were followed.

## Liver lipid analysis

Liver samples were homogenized in isopropanol (50 mg/mL). The tissue homogenate was centrifuged at 4°C and 10,000 rpm for 10 min. The organic phase of the supernatants was recovered, and triglyceride and total cholesterol levels were determined using commercial kits following the manufacturer's instructions (Cayman Chemical MT Item No. 10010303, Spinreact™ ref. no. 1001090).

## Histological score for MASLD

Kleiner's score was calculated by using a histological and semiquantitative scoring system for nonalcoholic fatty liver disease [26].

## Measurement of serum cytokines

The levels of IL-1β, TNF-α, IL-6, MCP-1, vascular endothelial growth factor (VEGF), IL-2, IFN-γ, and IL-10 were determined in serum samples using commercially available ELISA kits (PEPROTEC®).

## Cell culture and treatment

The human hepatoma-derived cell line Huh-7 was obtained from the American Type Culture Collection (ATCC, US). The cells were grown in high-glucose DMEM (Gibco, US) supplemented with 10% fetal bovine serum (Biowest) and 1% penicillin and streptomycin (Invitrogen, US) at 37°C under 5% $CO_2$ in a 95% humidified atmosphere. To induce FFA uptake, Huh-7 cells were exposed to FFAs (oleic acid and palmitic acid at a 4:1 volume ratio) for 24 h in the absence or presence of 1 mM IFC-305. Stock solutions of 30 mM FFAs were diluted at the indicated concentrations in serum-free DMEM supplemented with 1% FFA-free BSA as previously described [27].

## Oil red O staining and quantification

Huh-7 cells were cultured in 24-well plates at a density of 1.5 X $10^5$ cells/well and treated with different concentrations of FFAs and IFC-305 (1 mM) for 24 hours. The cells were washed two times with PBS and fixed in 4% buffered formalin for 15 min at room temperature. After fixation, the cells were washed twice with PBS and incubated with oil red O solution prepared as previously described for 1 hour at room temperature [28]. The stained cells were then washed in distilled water, isopropanol was added to each sample, and the samples were shaken at room temperature for 10 min. Then, the absorbance of the samples was read at 510 nm on a Cytation 3 Cell Imaging Multi-Mode Reader by BioTek Instruments, Inc., Vermont, U.S.A.

## Statistical analysis

The data are expressed as the mean ± standard error of the mean (SEM). Statistically significant differences among the experimental groups were determined by one-way analysis of variance (ANOVA) followed by Tukey's test for multiple comparisons (except in the weight gain graph, for which the applied statistic is described in the figure legend). Significance was set at p < 0.05. Graphs were created with GraphPad Prism software version 6.0.

# Results

## Experimental model characterization

We determined weight gain (growth curve), serum triglycerides, LDL cholesterol, serum insulin and glucose and insulin tolerance to characterize the experimental model used in this work (Fig 1). We evaluated the serum levels of AST and ALT in the different experimental groups. There was a tendency toward increased ALT in the HFS group; however, there were no statistically significant differences in the results obtained for both enzymes (S1 Fig). HFS and HFS +IFC groups showed a lower food intake but higher water + sucrose consumption than the rest of the groups (S1 Table). The results suggested that the IFC-305 administration effects are not directly related with a difference in the caloric intake. Sánchez-Tapia et al., determined that even though there are differences in the food and water consumption in rats fed with the high-fat and sucrose diet used in this model, compared with control rats, there are not changes in the caloric or energy intake among the groups [25].

HFS induced an increase of approximately 2-fold in weight compared to that in the control group (Fig 1A); treatment of the rats fed HFS with IFC-305 partially prevented the weight gain (HFS+IFC-305 group). Triglycerides, LDL-C and insulin levels were increased in the HFS group compared to the control group (Fig 1B–1D, respectively). In the HFS+IFC-305 group, a decrease in circulating triglycerides, a small but nonsignificant decrease in LDL cholesterol, and no differences in serum insulin concentration were observed; there was no effect of IFC-305 administration. Fig 1E shows the glucose tolerance curve. In the HFS group, there was a significant increase in circulating glucose 60 min after the administration of glucose solution until 120 min. Interestingly, in the HFS+IFC-305 group, glucose tolerance was equal to that in the control group, demonstrating that IFC-305 regulates glucose levels. This effect was quantified by the area under the curve (AUC) (Fig 1G). The IFC-305 compound did not induce changes in glucose levels in either curve under the same conditions (data not shown).

In addition, the effect of insulin administration (100 IU/ml) on circulating glucose is shown in Fig 1F. A significant increase in glucose level was observed 30 min after the administration of glucose solution in the HFS group compared to the control group, with a maximum level observed at 90 min, indicating insulin tolerance. Glucose levels diminished nearly to control-group levels by 120 min. There were no differences between the control and HFS+IFC-305 groups. This result was quantified according to the AUCs (Fig 1H).

## IFC-305 prevents histological alterations and hepatic lipid accumulation

Fig 2A shows images of the representative livers for each experimental group as well as their corresponding histological slices with hematoxylin and eosin staining. The images of the control group show the characteristics of a normal liver, such as its color and architecture, without the presence of an inflammatory infiltrate or accumulation of fat in the hepatocytes. In the whole liver picture in the HFS sample, a lighter and mottled color is observed compared to that in the control sample; moreover, an altered hepatic architecture and the presence of macrovesicular steatosis indicated by the  symbol are observed. A representative section of

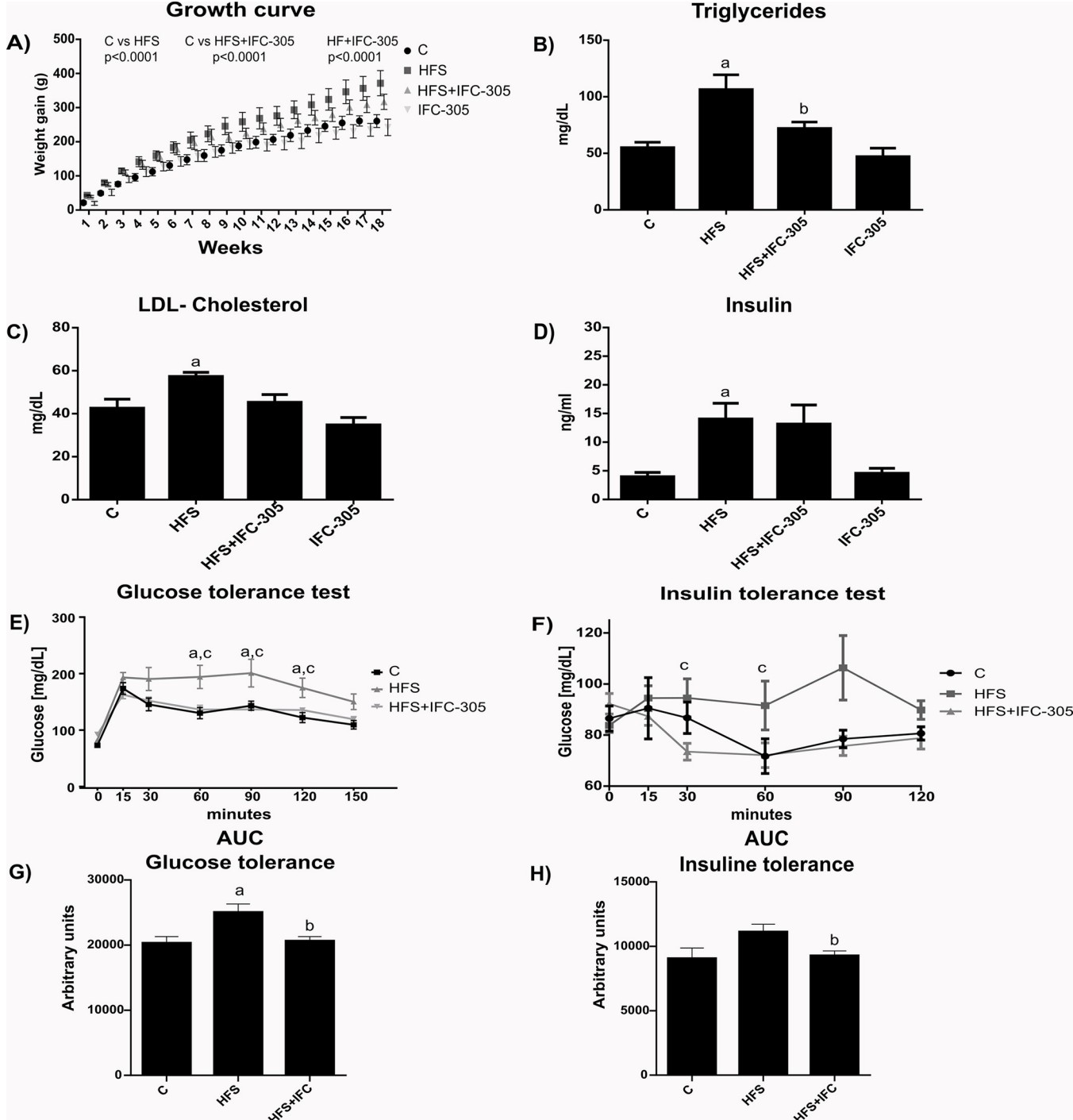

**Fig 1. Changes in body weight and elements associated with metabolic syndrome induced by HFS and its regulation by IFC-305.** (A) Weight gain was measured twice a week for 18 weeks. Statistical differences were analyzed with ANOVA followed by Fisher's test without corrections. Differences between groups were estimated considering each treatment mean value throughout time and were considered significant when p≤0.05. (B), Triglycerides, (C) LDL cholesterol, and (D) insulin were determined at the end of the treatments. (E) Glucose intolerance test. Blood glucose concentrations were measured 0, 15, 30, 60, 90, 120 and 150 minutes after oral glucose administration (3 g/kg). (F) Blood glucose concentrations 0, 15, 30, 60, 90 and 120 minutes after insulin administration (100 IU/ml). Glucose levels from the glucose and insulin tolerance tests were quantified as the mean from the area under the curve (AUC) of each animal (G and H, respectively). Differences were considered significant when p≤0.05. "a" indicates a significant difference compared to the control group; "b" indicates a significant difference compared to the HFS group; "c" indicates a significant difference compared to the HFS+IFC-305 group. The glucose levels from the glucose and insulin tolerance tests were quantified as the area under the curve (AUC) values.

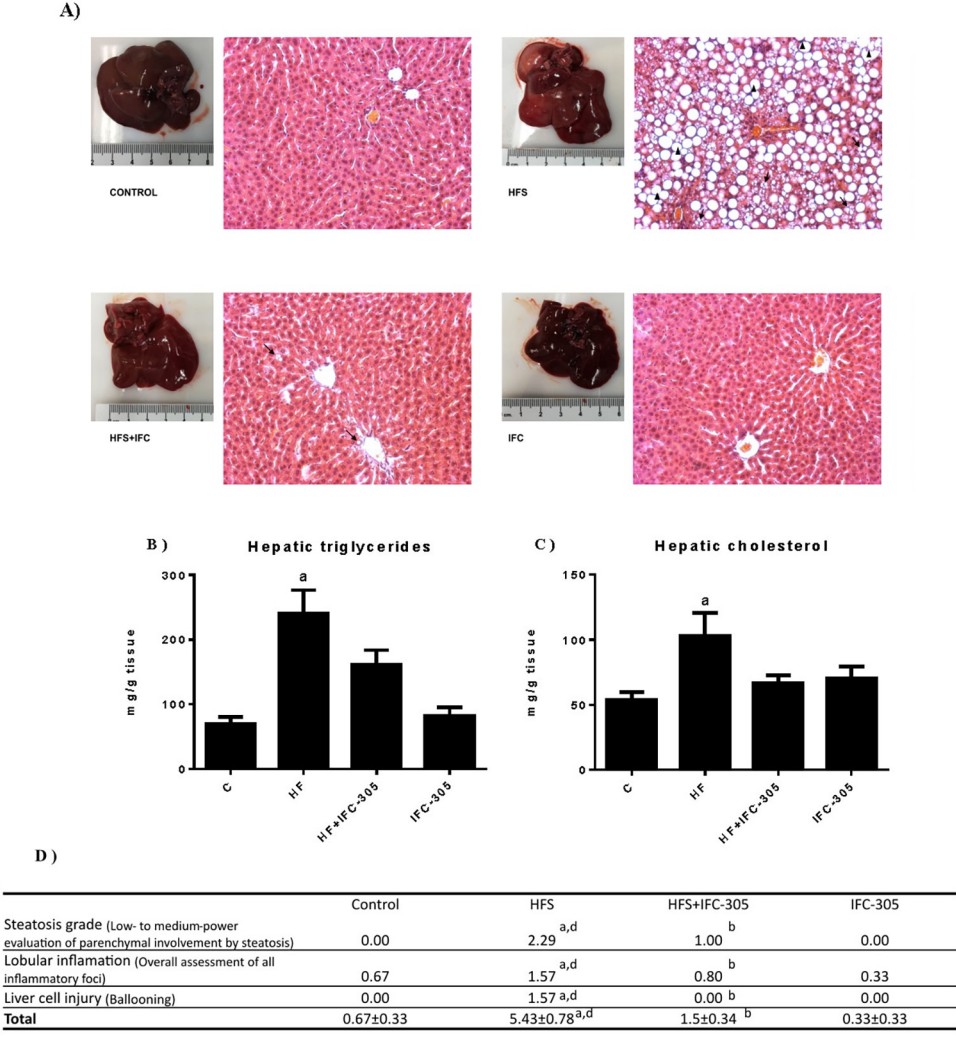

**Fig 2. IFC-305 prevents alterations in the hepatic parenchyma and the development of steatosis.** (A) Representative liver sections from each experimental group were stained with H&E and observed at 20X magnification. Macrovesicular and microvesicular steatosis are indicated by ↑ and ▲, respectively. (B) Hepatic triglyceride and (C) cholesterol content. (D) H&E-stained liver sections observed at 20X magnification were analyzed according to semiquantitative Kleiner's criteria. Statistically significant differences were determined by one-way analysis of variance (ANOVA) with multiple comparisons. "a" indicates a significant difference compared to the control group; "b" indicates a significant difference compared to the HFS group; "d" indicates a significant difference compared to the IFC group. A difference was considered significant when p≤0.05.

HFS+IFC-305 is shown in Fig 2C, which shows the ability of the compound to prevent macrovesicular steatosis; some lipid droplets can be observed, as indicated by arrowheads. No changes were induced by the administration of IFC-305 alone. Liver triglyceride and cholesterol levels were significantly increased in the HFS group compared to the control group, while the HFS+IFC-305 group showed no significant increase. The IFC-305 group showed that this compound had no effect on these parameters (Fig 2B and 2C). In clinical practice, the steatosis grade (from simple steatosis to MASH) can be determined by Kleiner's scoring system [26]. According to this score, the group fed an HFS diet showed a hepatic steatohepatitis phenotype, since liver samples had ≥34% parenchymal involvement by steatosis, as well as various inflammatory foci per X20 field and several cells with a ballooning phenotype due to fat

accumulation in the hepatocytes. IFC-305 administration prevented the progression of hepatic disease compared to that in the HFS group (Fig 2D).

The results demonstrate the ability of IFC-305 to prevent the increases in cholesterol and triglyceride levels, regulate glucose levels and prevent the accumulation of fat in hepatocytes. For this reason, we evaluated the levels of AMPK, a protein downstream of the adiponectin signaling pathway. The HFS group tended to have increased AMPK levels compared to the control group, and the HFS+IFC-305 group maintained AMPK at a level similar to that in the control group. No significant differences were found between the experimental groups (S2 Fig).

### Inflammation induced by HFS and the anti-inflammatory effect of IFC-305

We evaluated the inflammatory state because it is associated with metabolic syndrome and insulin resistance. As shown in Fig 3A–3C, IL-1β, TNF-α and IL-6 levels were increased in the HFS group, but coadministration of IFC-305 prevented this, indicating an anti-inflammatory effect. The chemokine MCP-1 favors the migration of inflammatory macrophages [29]. The level of this cytokine was significantly increased in response to HFS, but IFC-305 prevented this increase (Fig 3D). VEGF is highly expressed in adipose tissue; here, its expression was increased in the HFS group, but IFC-305 decreased it significantly (Fig 3E), showing antiadipogenic activity. The level of IFN-γ, an inflammatory cytokine that favors the differentiation of macrophages, was increased significantly in the HFS group, but IFC-305 treatment prevented this increase, thus decreasing the inflammatory response (Fig 3F). The cytokine IL-10, a typical anti-inflammatory cytokine, decreased with the HFS diet compared to the control diet (Fig 3G). In the HFS +IFC-305 group, the level of the anti-inflammatory cytokine IL-10 was maintained.

### Electron microscopy-revealed changes in liver cells induced by HFS and the protective effect of IFC-305

Fig 4A shows a control sample with a normal hepatocyte. It contains mitochondria (MT) associated with rough endoplasmic reticulum (RER). The square in the right inferior border shows a magnified view. Fig 4B shows electron microscopy of a damaged liver (HFS). The mitochondria do not have their typical morphology, and fewer associations are observed with the RER. The presence of ribosomes is reduced, and the number of mitochondria is lower than that in the control group. The number of lipid vesicles was reduced in rats treated with IFC-305 (HFS +IFC-305), corroborating the findings of H&E staining; in addition, the loss of mitochondria was prevented, and the association of mitochondria with the RER was maintained (Fig 4C). Treatment with IFC-305 in rats fed a standard diet did not cause any phenotypic alterations in liver cells (Fig 4D).

### *In vitro* uptake of fatty acids in Huh-7 cells

To corroborate the effect observed in hepatic lipid content, we decided to evaluate the probable inhibitory effect of the IFC-305 compound on fatty acid uptake *in vitro*. Fig 5 shows the uptake of fatty acids in Huh-7 cells. Incubation with IFC-305 for 24 h partially inhibited fatty acid uptake measured using the oil red method. These results were in accordance with the findings of the lipid content in the liver from the HFS rats shown histologically and by assessment of the hepatic triglyceride and cholesterol content.

### Discussion

The compound IFC-305 is an adenosine derivative that has been shown to have the same hepatoprotective effects as adenosine (200 mg/kg) at a dose 4 times lower (50 mg/kg) in addition to

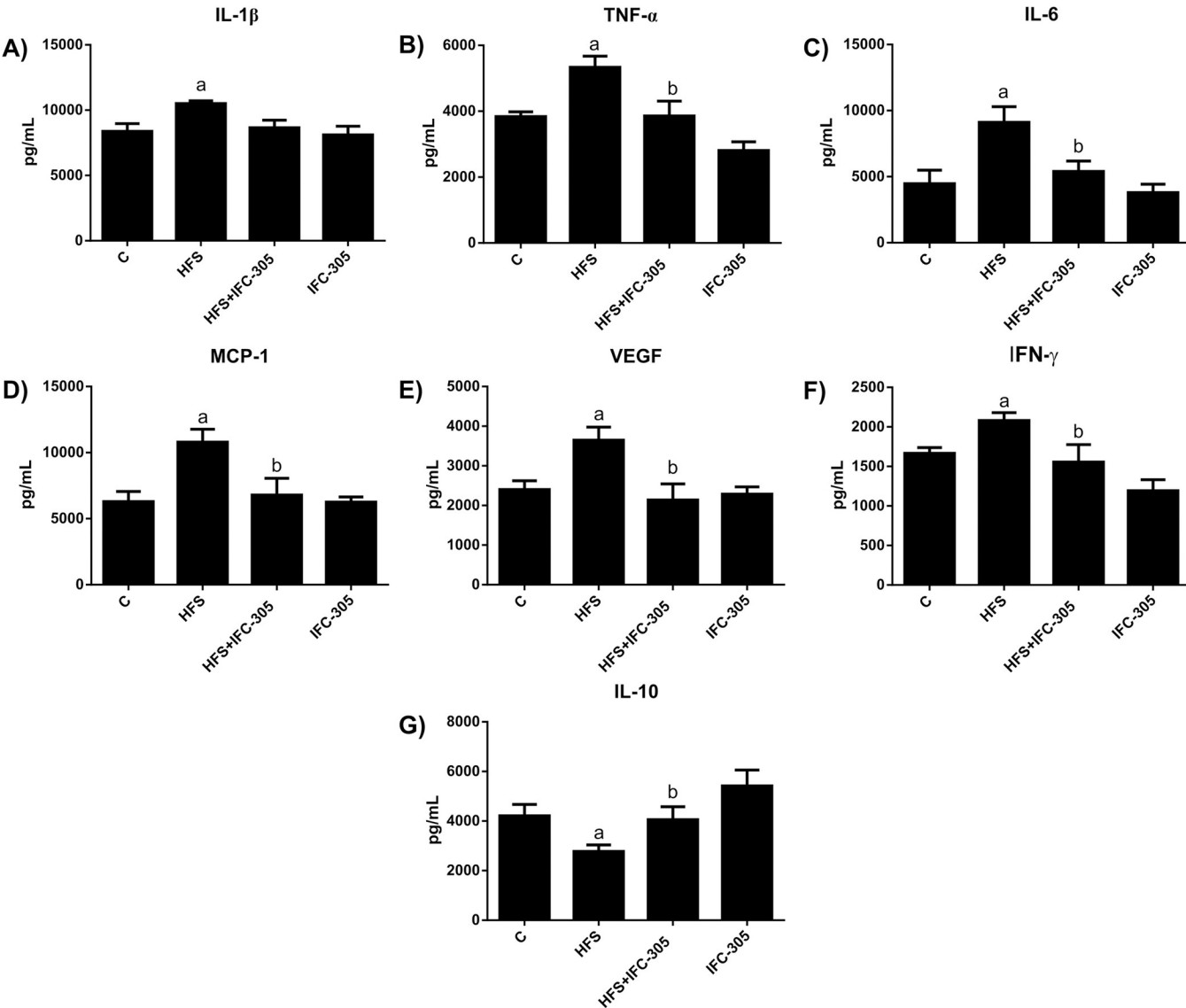

**Fig 3. Inflammation induced by a high-fat diet and sucrose was inhibited by IFC-305.** Determination of serum cytokines. The levels of the proinflammatory cytokines IL-1β, TNF-α, IL-6, and IFN-γ; the macrophage-recruiting chemokines MCP-1 and VEGF; and the anti-inflammatory interleukin IL-10 were measured. Statistically significant differences were determined by one-way analysis of variance (ANOVA) with multiple comparisons. "a" indicates a statistically significant difference compared to the control group; "b" indicates a significant difference compared to the HFS group. Differences were considered significant when P≤0.05.

exhibited a longer half-life than adenosine. Some of the observed effects of IFC-305 include decreased collagen levels, increased collagenolytic activity, and increased DNA synthesis as well as recovery of liver function in a CCl₄-induced cirrhosis model [16]. In addition, an evaluation of global DNA methylation has revealed decreases in methylation and demethylation and recovery in the presence of IFC-305 in cirrhosis, indicating a double increase in global DNA methylation [30]. In *in vitro* studies using isolated hepatic stellate cells, IFC-305 has been found to be able to inhibit the TGF-β signaling pathway that induces collagen gene expression (data not published). In a model of hepatocellular carcinoma induced by DEN, IFC-305 has been demonstrated to have chemoprotective effects, having an important effect on the recovery of mitochondrial function [20, 21]. Moreover, previous findings on the hepatoprotective

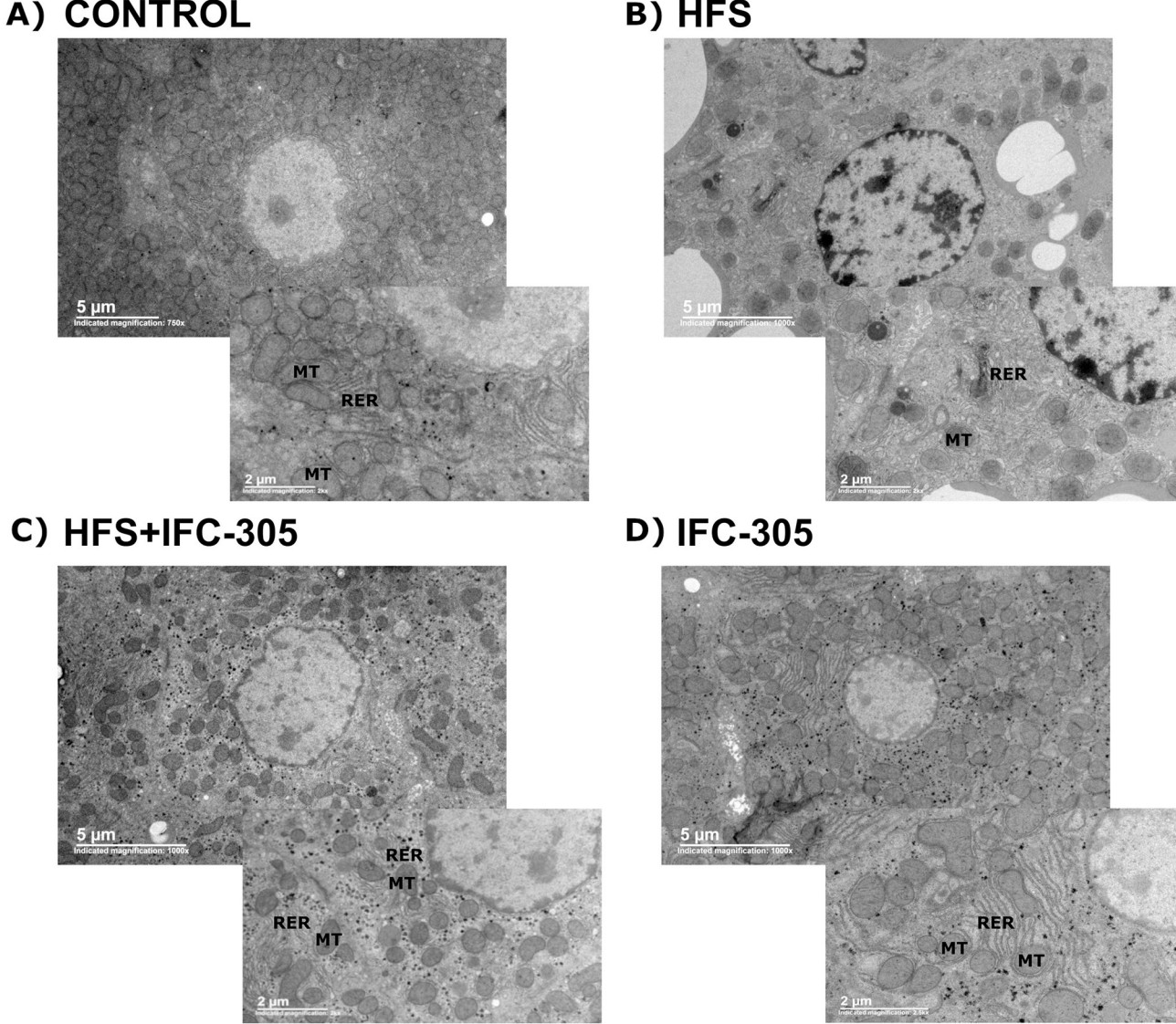

**Fig 4. Morphological changes evaluated by electron microscopy.** Each panel shows representative samples from the control, HFS, HFS+IFC-305 and IFC-305 groups. Each group was observed at different magnifications. MT: mitochondria, RER: rough endoplasmic reticulum.

role of IFC-305 and its inhibitory effects on the development of fatty liver prompted us to evaluate the effect of this compound in an experimental model induced by HFS for 18 weeks. The results suggest that IFC-305 is a promising molecule for inhibiting the establishment of steatosis and regulating glucose levels, systemic inflammation and the development of metabolic syndrome.

It has been reported that there is no precise correlation between the serum level of the ALT enzyme and fatty liver; however, ALT has also been suggested as a possible marker [31]. On the other hand, a relationship between AST and MASLD has not been reported, unlike the case for liver damage associated with alcohol consumption, in which an increase in the AST: ALT ratio is observed [32]. The determination of these parameters in our metabolic syndrome model showed only a slight increase in the ALT level that was not observed in the HF+IFC-305 group, and there was no difference in the AST level in the experimental groups, suggesting a protective effect against the accumulation of fat in the liver.

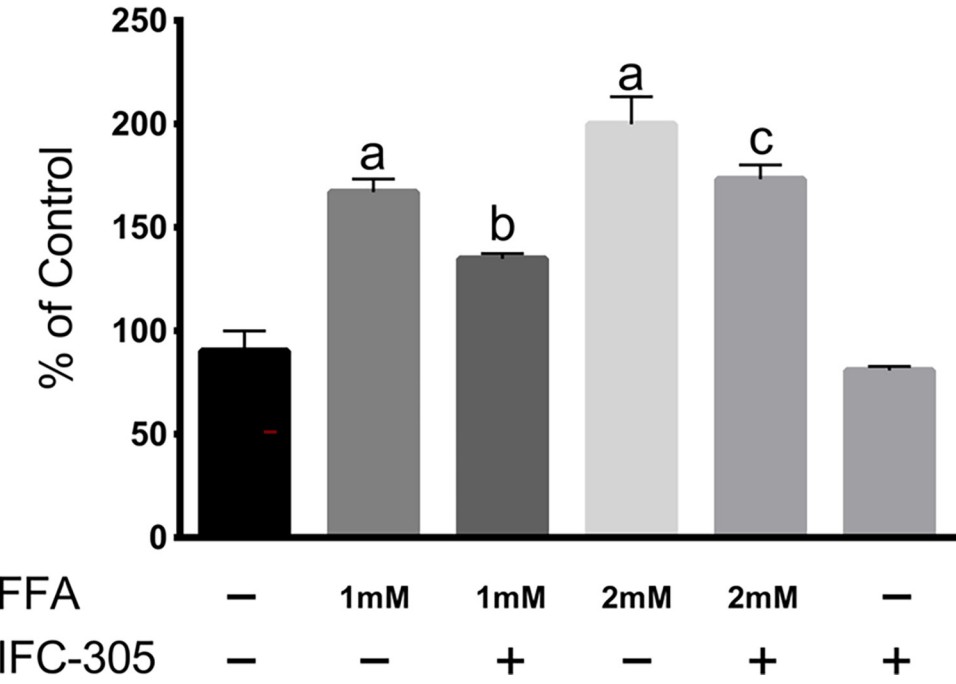

**Fig 5. IFC-305 partially prevented fatty acid uptake by Huh-7 cells.** Huh-7 cells were treated with the indicated concentrations of FFAs and IFC-305 for 24 h and then stained with oil red reagent, and the absorbance was read at 510 nm. All results are shown as the mean ± SD (n = 3) from a representative experiment. Statistically significant differences were determined by one-way analysis of variance (ANOVA) with multiple comparisons. "a" indicates statistically significant differences from the control group, P < 0.01. "b" indicates statistically significant differences from the 1 mM FFA group, P < 0.01. "c" indicates statistically significant differences from the 2 mM FFA group, P < 0.01.

The growth curve showed weight gain that was partially prevented by IFC-305 administration. In addition, this compound had beneficial effects on triglyceride levels and somewhat beneficial effects on insulin and LDL cholesterol levels. Previous studies have demonstrated the ability of adenosine, the base molecule of IFC-305, to diminish hyperlipidemia in stroke-prone spontaneously hypertensive rats fed a high-fat diet by downregulating the A2B receptor [15].

In this study, the animals fed the HFS diet developed glucose intolerance without developing diabetes. After fasting at time 0, they did not exhibit differences from the control animals, but when exogenous glucose was administered, their levels remained elevated until 120 min. The observed insulin resistance in the HFS group indicates the inability of insulin to increase glucose uptake [33]. Here, animals fed HFS were unable to maintain normal glucose levels, so they maintained high levels until 90 min after glucose intake; this result was directly related to the loss of sensitivity to insulin signaling. Some nucleosides, such as adenosine, have been shown to exert hypoglycemic effects during the postprandial period [14]. The IFC-305 compound demonstrated a similar ability and therefore regulated serum insulin levels probably in the same way as adenosine, which could function as an integral treatment because of its different metabolic effects. For instance, adenosine regulates lipolysis and lipogenesis through the A1 receptor. Additionally, its insulin-like functions are well described [34]. Moreover, this molecule has ubiquitous cell signaling actions on energy homeostasis [35].

In this work, animals fed HFS developed liver damage, which was observed macro- and microscopically. Intrahepatic lipid accumulation results from lipolysis, FFA uptake and very low-density-lipoprotein (VLDL) synthesis as well as reduced FFA oxidation and triglyceride (TG) export; moreover, these processes are frequently associated with the inflammatory

response [36]. Adenosine has previously been shown to prevent and reverse fatty liver induced by ethanol administration by increasing ATP levels and stimulating oxidation in mitochondria [37]. Histologically, a large number of lipid vacuoles and the phenomenon of "ballooning" were observed, which has been proposed as a histopathological characteristic of NASH [38]. In this work, this characteristic was observed in the HFS group. In contrast, treatment with IFC-305 significantly prevented the accumulation of lipids in hepatocytes, notably improving macroscopic liver appearance. The cholesterol and triglyceride levels observed in the HFS+IFC-305 group corroborated these results. The assay of fatty acid uptake by Huh-7 cells enabled us to confirm the ability of IFC-305 to reduce the accumulation of lipids in hepatocytes, explaining in part the findings *in vivo*. Previously, it has been shown that adenosine prevents approximately 50% of the accumulation of triglycerides in fatty liver induced by ethanol [37]. The levels of p-AMPK observed in the HFS group and in the HFS+IFC-305 group suggest that the mechanism of action is not through induction of the adiponectin signaling pathway; however, further research is needed. Interestingly, p-AMPK is related to other cellular processes, such as autophagy, a cellular recycling process. If phosphorylation of AMPK is not induced, AMPK does not activate mTOR, an inhibitor of autophagy. We previously demonstrated that IFC-305 induces autophagy and that this cellular process can maintain lipid droplet homeostasis [39]. In addition, IFC-305 favors mitophagic flux, stimulating mitochondrial activity and oxidative metabolism [21, 22]. Moreover, in one study, adenosine prevented free fatty acid hepatic oxidation by inhibiting extramitochondrial acyl-CoA synthetase, thus diminishing plasma fatty acid uptake [40]; however, this effect has since been considered to be transient, since in another study, it was found that adenosine decreases the generation of ketone bodies in a model of fatty liver induced by cycloheximide, suggesting its ability to induce β-oxidation [18]. Once in hepatocytes, fatty acids that were released from adipocytes need to be activated as acyl-CoA to be metabolized, either via β-oxidation in mitochondria or by being used in TG, phospholipid or cholesterol synthesis. The family of enzymes responsible for the activation of fatty acids are called long-chain acyl-CoA synthetases (ACSLs). In this study, we determined the expression of the ACSL1 and ACSL3 isoforms in an *in vitro* model of fatty acid incorporation in HepG2 cells. We found that the expression of the ACSL3 isoform was decreased in cells exposed to FFAs compared to control cells, while those incubated with IFC-305 in the presence of FFAs or alone had recovery of ACSL3 expression. Regarding the ACSL1 isoform, no differences were observed, although there was a tendency toward an increase in cells incubated with IFC-305. One possibility is that the slight increases in the expression of ACSL1 and ACSL3 induced by IFC-305 favor the synthesis of acyl-CoA for oxidation and not for the synthesis of TG, since in both the *in vivo* and *in vitro* models, a decrease in the accumulation of TG deposits was observed. This is in accordance with another study in which the expression of ACSL3 was induced with oncostatin M in a model of hyperlipidemic hamsters and in HepG2 cells; in that study, greater oxidation of FFAs and a decrease in the accumulation of TG were observed [41]. In another study, in hamsters fed a high-fructose diet, the expression of hepatic ACSL3 was decreased, and re-expression of ACSL3 was accompanied by a decrease in the TG level in the liver, similar to the findings with IFC-305 [42].

According to this, MASLD, obesity and insulin resistance play important roles in lipid disorders, and steatosis has been correlated with insulin resistance [36].

Mitochondria have been reported to undergo structural and functional alterations during the progression of liver disease associated with insulin resistance. Observed changes include strong condensation of structures of the mitochondrial matrix or changes in the matrix density, loss of ridges and a decrease in mitochondrial biogenesis [43, 44]. Liver sections were also analyzed by electron microscopy, which revealed diminished and morphologically altered mitochondria in the animals fed the high-fat diet and 10% sucrose in drinking water. In the

group fed the HFS diet supplemented with sucrose, the simultaneous administration of IFC-305 prevented the mitochondrial alterations observed by electron microscopy, as previously demonstrated in hepatocarcinoma [21].

HFS induced systemic inflammation, and the anti-inflammatory effect of IFC-305 has been demonstrated in a previous study on a cirrhosis model. In that study, IFC-305 reduced IL-1β and IL-6 levels and increased IL-10 levels [45]. IL-1β, TNF-α and IL-6 are inflammatory cytokines correlated with and involved in obesity-related inflammation and insulin resistance, respectively [29, 46]. As we mentioned earlier, systemic inflammation is directly related to metabolic syndrome and may promote NASH. During chronic inflammation, recruitment of M1 macrophages is promoted, and TNF-α and MCP-1 are hypersecreted in both adipose tissue and hepatocytes, which contribute to the establishment of cellular infiltration. In patients with metabolic syndrome, increased serum levels of IL-6 and TNF-α have been found to be associated with the severity of metabolic syndrome development [47]. Previous research has demonstrated an increase in the number of M1 macrophages in an experimental model of cirrhosis induced by $CCl_4$; in that research, treatment with IFC-305 prevented this increase and favored the production of the anti-inflammatory cytokine IL-10 [45]. According to these findings and the results obtained here regarding the ability of IFC-305 to prevent the increases in the levels of proinflammatory cytokines and a decrease in Kleiner's score, which reflects lobular inflammation in metabolic syndrome, we propose that this effect is related to a regulation of immune cells. Adipose tissue requires adequate vasculature, and production of the angiogenic factor VEGF is important, but an increase in the VEGF level is a reflection of an increase in the number of adipocytes and therefore obesity [48]. VEGF is proposed to play an important role in weight gain, and a high concentration of this growth factor in serum is positively correlated with increases in body mass index and visceral fat deposition; in contrast, in adipose tissue, VEGF overexpression promotes angiogenesis. Likewise, VEGF dysregulation may be a risk factor for vascular diseases [49].

Altogether, the results presented here demonstrate the beneficial effects of IFC-305 in the experimental model in male rats. However, further studies are required to elucidate the effects with consideration of more variables, since some sex differences have been found in the establishment and progression of MASLD. Such studies will enable us to understand the effects and ensure that they are the same in both sexes [50].

These results demonstrate that IFC-305 prevents the alterations associated with the progression of metabolic syndrome and liver damage associated with excessive fat and sugar consumption in an experimental model. These effects result from the regulation of blood glucose and other serum elements, the preservation of liver structure and the prevention of inflammation. The results indicate that IFC-305 may be a promising molecule for the treatment of metabolic syndrome and related diseases.

## Supporting information

**S1 Fig. Determination of liver damage by enzymatic transaminases activities.** (A) AST and (B) ALT activity was determined in serum from rats in the control (n = 3), HFS (n = 10), and HFS+IFC-305 (n = 10) groups. The values represent the mean of experiments performed in duplicate assays ± SEM.
(TIFF)

**S2 Fig. pAMPK protein level.** pAMPK protein levels in samples of liver tissue determined by Western blot analysis from rats in the control (n = 3), HFS (n = 4), HFS+IFC-305 (n = 4), and IFC-305 (n = 4) groups. GAPDH was used as an internal control. A) The signal intensities were determined by densitometric analysis of treated blots, and the values were calculated as

the ratio of p-AMPK to GAPDH. Each bar represents the mean value of experiments ± SEM.
B) Representative blot for each group.
(TIFF)

**S3 Fig. mRNA expression of ACSL1 and ACSL3 in HepG2 cells.** mRNA expression of
ACSL1 and ACSL3 in HepG2 cells. HepG2 cells were cultured in serum-free DMEM supple-
mented with 1% BSA-free FFAs (control) in the presence of 1 mM IFC-305 or FFA with or
without 1 mM IFC-305 for 24 h. Real-time quantitative polymerase chain reaction (qPCR)
analysis of the mRNA expression of (A) ACSL1 and (B) ACSL3 isoforms normalized to β-
actin. The relative mRNA levels were calculated using the comparative ΔΔCt method. mRNA
expression is expressed as the mean value ± SEM from three independent experiments.
**P $< 0.001$.
(TIFF)

**S1 Table. Food and water consumption.** Food and water intake (or water+sucrose solution
for HFS and HFS+IFC groups) were measured twice a week. Captured data are expressed as
mean values ± SEM from 0, 8, 12 and 18 weeks of treatment. Statistical differences were deter-
mined with ANOVA followed by a multiple comparison test where "a" indicates a significant
difference compared to the control group, P≤0.05.
(DOCX)

**S1 Data.**
(ZIP)

## Acknowledgments

We would like to thank Dr. Rodolfo Paredes Díaz for their support at the imaging unit of the
institute; Claudia Rivera Cerecedo, DVM, and Gabriela Xóchitl Ayala Méndez, MSc., from the
Bioterium for providing the rats; Elizabeth Morales Sánchez, MSc., Sandra Rodríguez Mon-
taño., MSc., and Leticia Ramírez Lugo, PhD., from the Histology Unit; and Mónica Sánchez-
Tapia, PhD, for her support in preparing the diets.

## Author Contributions

**Conceptualization:** Victoria Chagoya-de Sánchez.

**Data curation:** Enrique Chávez, Alejandro Rusbel Aparicio-Cadena, Victoria Chagoya-de
Sánchez.

**Formal analysis:** Enrique Chávez, Alejandro Rusbel Aparicio-Cadena, Gabriela
Velasco-Loyden, María Guadalupe Lozano-Rosas, Mariana Domínguez-López, Amairani
Cancino-Bello, Alejandro Cabrera-Aguilar.

**Funding acquisition:** Enrique Chávez, Victoria Chagoya-de Sánchez.

**Investigation:** Victoria Chagoya-de Sánchez.

**Methodology:** Enrique Chávez, Alejandro Rusbel Aparicio-Cadena, Gabriela Velasco-Loyden,
María Guadalupe Lozano-Rosas, Mariana Domínguez-López, Amairani Cancino-Bello,
Nimbe Torres, Armando R. Tovar, Alejandro Cabrera-Aguilar.

**Supervision:** Enrique Chávez, Gabriela Velasco-Loyden, Victoria Chagoya-de Sánchez.

**Validation:** Nimbe Torres, Armando R. Tovar, Victoria Chagoya-de Sánchez.

**Writing – original draft:** Enrique Chávez, Alejandro Rusbel Aparicio-Cadena, Gabriela Velasco-Loyden, María Guadalupe Lozano-Rosas, Mariana Domínguez-López.

**Writing – review & editing:** Enrique Chávez, Alejandro Rusbel Aparicio-Cadena.

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
