## [Decision Letter · Decision Letter 0]

3 Jul 2023

PONE-D-23-08771An adenosine derivative prevents the alterations observed in metabolic syndrome in a rat model induced by a rich high-fat diet and sucrose supplementationPLOS ONE

Dear Dr. Chagoya de Sánchez,

Thank you for submitting your manuscript to PLOS ONE. After careful consideration, we feel that it has merit but does not fully meet PLOS ONE’s publication criteria as it currently stands. Therefore, we invite you to submit a revised version of the manuscript that addresses the points raised during the review process.

We look forward to receiving your revised manuscript.

Kind regards,

Nobuyuki Takahashi, Ph.D.

Academic Editor

PLOS ONE

Journal Requirements:

3. Please expand the acronym “CONACYT”, "PAPIIT-UNAM" (as indicated in your financial disclosure) so that it states the name of your funders in full.

"ARAC is recipient of a CONACYT-Mexico  fellowship 720321.

VCdS received financial support from PAPIIT-UNAM IN214419. "

Additional Editor Comments:

I apologize for the delay in the reviewing process of your manuscript. This is because enough reviewers cannot be found. The next revision process will not take as long. Please carefully consider the reviewers' comments and revise your manuscript.

Reviewers' comments:

Reviewer's Responses to Questions

**Comments to the Author**

1. Is the manuscript technically sound, and do the data support the conclusions?

Reviewer #1: Yes

Reviewer #2: Partly

2. Has the statistical analysis been performed appropriately and rigorously? 

Reviewer #1: I Don't Know

Reviewer #2: Yes

3. Have the authors made all data underlying the findings in their manuscript fully available?

Reviewer #1: No

Reviewer #2: Yes

4. Is the manuscript presented in an intelligible fashion and written in standard English?

Reviewer #1: No

Reviewer #2: No

5. Review Comments to the Author

Reviewer #1: In this paper, the authors examined effects of IFC-305, an adenosine derivative, on metabolic syndrome and nonalcoholic fatty liver disease (NAFLD) using an animal model and cell line. As a result, IFC-305 ameliorated obesity, serum levels of triglyceride and proinflammatory cytokines, glucose tolerance, and hepatic steatosis. The results of the present study are interesting, but the authors should consider the following points.

Major points:

1. The authors should examine serum levels of aspartate aminotransferase (AST) and alanine aminotransferase (ALT) to confirm the effects of IFC-305 on hepatocellular injury. Serum levels of adiponectin should also be examined

2. On histopathological evaluation of NAFLD, not only steatosis but also lobular inflammation, hepatocellular ballooning, and fibrosis are important. The authors should evaluate these items semi-quantitatively using Kleiner’s scoring system (Hepatology 2005; 41: 1313-21).

3. The quality of English is insufficient. The authors should seek assistance of a native English speaker.

Minor points

1. Age of the rats at initiation of the experiment should be described.

2. Data of food consumption and calorie intake should be shown if the authors checked them.

3. Line 113: “HFC” should be revised as “HFS.”

4. Line 171, 277, Fig. 3: “INF-γ” should be revised as “IFN-γ.”

5. Line 234: “glucose” should be revised as “insulin.”

6. Line 237: “de” should be revised as “the.”

7. Line 283: “Microscopic” should be revised as “Electron-microscopic.”

8. Line 286: “microscopy” should be revised as “electron microscopy.”

9. Line 304: “red oil” should be revised as “oil red.”

10. Line 347-8: The authors describe that IFC-305 had beneficial effects on insulin and LDL cholesterol levels, but there are no significant differences between the HFS and HFS+IFC-305 groups in Fig. 1(C,D).

11. Line 410: “NAFD” should be revised as “NAFLD.”

12. Fig. 2 (B,C): It seems that there are remarkable differences between the HFS and HFS+IFC-305 groups in the hepatic triglyceride and cholesterol levels. Aren’t the differences statistically significant?

13. Legend of Fig. 4: “N”, “L”, and “GA” are not used in the figure. “MT” and “RER” are defined two times.

Reviewer #2: The authors demonstrated here that an adenosine derivative, IFC-305, ameliorated high-fat-diet-induced hepatic inflammation and steatosis. These findings are very interesting. However, the data presented in this study are only partially convincing. Additional experimental data are needed to confirm the authors' conclusion, as the follows.

1) The authors should examine hepatic macrophage markers such as mRNA levels and/or immunohistochemistry of F4/80. According to the data shown in Fig.3, M1 macrophages would be decreased and/or inactivated. This should be investigated to confirm the authors' conclusion.

2) The authors should measure the mRNA expression levels of lipid metabolism in the liver and Huh-7 hepatocytes. In general, amelioration of inflammation improves the dysfunction of lipid metabolism in the liver. Therefore, it is necessary to examine the mRNA expression levels of fatty acid oxidation and synthesis in the liver of the IFC-305-fed mice and the IFC-305-treated Huh-7 hepatocytes.

3) The authors should measure serum levels of the liver injury markers such as AST and ALT.

4) The authors should check the minor points and English throughout the manuscript. There are many errors.

6. PLOS authors have the option to publish the peer review history of their article (what does this mean?). If published, this will include your full peer review and any attached files.

Reviewer #1: No

Reviewer #2: No

---

## [Author Response · Author response to Decision Letter 0]

4 Sep 2023

We really appreciate all the comments you made about the manuscript we submitted to be considered for publication in Plos One.

All the comments you recommended were considered in order to improve the manuscript.

Reviewer #1: In this paper, the authors examined effects of IFC-305, an adenosine derivative, on metabolic syndrome and nonalcoholic fatty liver disease (NAFLD) using an animal model and cell line. As a result, IFC-305 ameliorated obesity, serum levels of triglyceride and proinflammatory cytokines, glucose tolerance, and hepatic steatosis. The results of the present study are interesting, but the authors should consider the following points.

Major points:

1. The authors should examine serum levels of aspartate aminotransferase (AST) and alanine aminotransferase (ALT) to confirm the effects of IFC-305 on hepatocellular injury. Serum levels of adiponectin should also be examined

We evaluated the serum levels of AST and ALT as you recommended. There was not statistical significant difference in the results obtained, because of this,we decided to show them as a supplementary figure (S1 Fig).

The phollowing paragraph was added in the Results section

(Line 186-188) We evaluated the serum levels of AST and ALT in the different experimental groups. There was a tendency toward increased ALT in the HFS group; however, there were no statistically significant differences in the results obtained for both enzymes (S1 Fig). 

The phollowing paragraph was added in the Discussion section

(Line 336-343) It has been reported that there is no precise correlation between the serum level of the ALT enzyme and fatty liver; however, ALT has also been suggested as a possible marker (33). On the other hand, a relationship between AST and MASLD has not been reported, unlike the case for liver damage associated with alcohol consumption, in which an increase in the AST:ALT ratio is observed (34). The determination of these parameters in our metabolic syndrome model showed only a slight increase in the ALT level that was not observed in the HF+IFC-305 group, and there was no difference in the AST level in the experimental groups, suggesting a protective effect against the accumulation of fat in the liver.

Adiponectin is an important factor that stimulates fatty acid oxidation, inhibits glucose production, and has an anti-inflammatory effect. We thought it was a good idea to determine their levels in the different experimental groups as you suggested. However, due to time constraints, it was not possible for us to acquire the necessary supplies, since we did not have them and we did not want to delay sending the corrected version of the manuscript.

Despite this, we decided to determine by Western blot the levels of p-AMPK, a protein that is found downstream of the adiponectin signaling pathway and that is important for its response. The results are shown in S2 Fig.

The phollowing paragraphs was added in the Results section:

(Line 261-266) The results demonstrate the ability of IFC-305 to prevent the increases in cholesterol and triglyceride levels, regulate glucose levels and prevent the accumulation of fat in hepatocytes. For this reason, we evaluated the levels of AMPK, a protein downstream of the adiponectin signaling pathway. The HFS group tended to have increased AMPK levels compared to the control group, and the HFS+IFC-305 group maintained AMPK at a level similar to that in the control group. No significant differences were found between the experimental groups (S2 Fig).

The phollowing paragraphs was added in the Discussion section

(Line 367-368) Adenosine has previously been shown to prevent and reverse fatty liver induced by ethanol administration by increasing ATP levels and stimulating oxidation in mitochondria (39). 

(Line 377-384) The levels of p-AMPK observed in the HFS group and in the HFS+IFC-305 group suggest that the mechanism of action is not through induction of the adiponectin signaling pathway; however, further research is needed. Interestingly, p-AMPK is related to other cellular processes, such as autophagy, a cellular recycling process. If phosphorylation of AMPK is not induced, AMPK does not activate mTOR, an inhibitor of autophagy. We previously demonstrated that IFC-305 induces autophagy and that this cellular process can maintain lipid droplet homeostasis (41). In addition, IFC-305 favors mitophagic flux, stimulating mitochondrial activity and oxidative metabolism (22,42).

2. On histopathological evaluation of NAFLD, not only steatosis but also lobular inflammation, hepatocellular ballooning, and fibrosis are important. The authors should evaluate these items semi-quantitatively using Kleiner’s scoring system (Hepatology 2005; 41: 1313-21).

The semi-quantitative Kleiner’s score was analyzed according to the reference you recommended.

The following section was added to Methods:

(Line 149-151) Histological score for MASLD

Kleiner’s score was calculated by using a histological and semiquantitative scoring system for nonalcoholic fatty liver disease (27).Panel D has been added to Figure 4 showing the Kleiner’ score.

In addition, the following modifications were made to the manuscript:

- The phollowing sentence was added to the Legend of Fig. 2: 

(D) H&E stained liver sections observed at 20X magnification were analyzed according with semiquantitative Kleiner’s criteria.

In the Results section, the phollowing paragraph was added:

(243-249) In clinical practice, the steatosis grade (from simple steatosis to MASH) can be determined by Kleiner’s scoring system (27). According to this score, the group fed an HFS diet showed a hepatic steatohepatitis phenotype, since liver samples had ≥34% parenchymal involvement by steatosis, as well as various inflammatory foci per X20 field and several cells with a ballooning phenotype due to fat accumulation in the hepatocytes. IFC-305 administration prevented the progression of hepatic disease compared to that in the HFS group (Fig 2D).

3. The quality of English is insufficient. The authors should seek assistance of a native English speaker.

The corrected versión of the manuscript was sent to AJE.com for English style correction.

Minor points

1. Age of the rats at initiation of the experiment should be described.

The phrase (7 to 8 weeks old) was added in parentheses in the first line of the Animals section.

2. Data of food consumption and calorie intake should be shown if the authors checked them.

In the following table we show the results of the food and water consumption during the time that the experimental model lasted. We include the results as supplementary material (S1 Tab).

The following paragraph was added to the Discussion section 

(Line 188-194) HFS and HFS+IFC groups showed a lower food intake but higher water + sucrose consumption than the rest of the groups (S1 Tab). The results suggested that the IFC-305 administration effects are not directly related with a difference in the caloric intake. Sánchez-Tapia et al., determined that even though there are differences in the food and water consumption in rats fed with the high-fat and sucrose diet used in this model, compared with control rats, there are not changes in the caloric or energy intake among the groups (30).

3. Line 113: “HFC” should be revised as “HFS.”

HFC was changed by HFS.

4. Line 171, 277, Fig. 3: “INF-γ” should be revised as “IFN-γ.”

INF-γ was changed by IFN-γ.

5. Line 234: “glucose” should be revised as “insulin.” 

The term “glucose” is correct because the way to measure insulin tolerance is to observe how glucose levels behave over a period of time after insulin administration, as we described in the paragraph.

6. Line 237: “de” should be revised as “the.”

The word “de” was changed by “the”.

7. Line 283: “Microscopic” should be revised as “Electron-microscopic.”

“Microscopic” was changed by “Electron-microscopic”

8. Line 286: “microscopy” should be revised as “electron microscopy.”

“Microscopy” was changed by “electron-microscopy”

9. Line 304: “red oil” should be revised as “oil red.”

“red oil” was changed by “oil red”

10. Line 347-8: The authors describe that IFC-305 had beneficial effects on insulin and LDL cholesterol levels, but there are no significant differences between the HFS and HFS+IFC-305 groups in Fig. 1(C,D).

The phrase “on insulin, triglyceride and LDL cholesterol levels” was changed by “and partially on insulin and LDL”

11. Line 410: “NAFD” should be revised as “NAFLD.”

The acronym NAFLD was changed in the manuscript by MASLD (metabolic dysfunction–associated steatotic liver disease) according to new nomenclature adopted and published in Hepatology. 2023 Jun 24. doi: 10.1097/HEP.0000000000000520.

12. Fig. 2 (B,C): It seems that there are remarkable differences between the HFS and HFS+IFC-305 groups in the hepatic triglyceride and cholesterol levels. Aren’t the differences statistically significant?

According to the Tukey’s test for multiple comparisons, as we describe in the statistical analysis in the Method’s section, there is difference statistically significant just in the control group versus the HFS group in both parameters, triglyceride and cholesterol content. This allows us to suggest only a tendency to prevent the accumulation of triglycerides and cholesterol because of the treatment with IFC-305 in fed rats with the HFS diet.

13. Legend of Fig. 4: “N”, “L”, and “GA” are not used in the figure. “MT” and “RER” are defined two times.

“N”, “L”, and “GA” were deleted from the legend of Fig. 4. The repited “MT” and “RER” definition were removed.

Reviewer #2: The authors demonstrated here that an adenosine derivative, IFC-305, ameliorated high-fat-diet-induced hepatic inflammation and steatosis. These findings are very interesting. However, the data presented in this study are only partially convincing. Additional experimental data are needed to confirm the authors' conclusion, as the follows.

1) The authors should examine hepatic macrophage markers such as mRNA levels and/or immunohistochemistry of F4/80. According to the data shown in Fig.3, M1 macrophages would be decreased and/or inactivated. This should be investigated to confirm the authors' conclusion.

It is known that M1 macrophages are increased in metabolic syndrome (DOI: 10.1042/BJ20111708), and as we mentioned in the manuscript according to the cytokine profile, it is suggested that M1 macrophages decreased with the IFC-305 treatment. We agree that exploring how these macrophages behave could provide more detailed information on the effect of IFC-305 on the immune response in this experimental model, and could provide more precise information about if it acts directly limiting this cell population for the development and recovery of the liver.

Unfortunately at this moment we do not have available the F4/80 antibody for immunohistochemistry and it takes approximately 2 months to arrive.

Due to the anti-inflammatory effects of IFC-305 previously resported and that we briefly describe below, and the fact that we evaluate lobular inflammation, hepatocellular ballooning, and fibrosis by using Kleiner’s scoring system (Hepatology 2005; 41: 1313-21), we believe that this information could support our conclusions.

Panel D has been added to Figure 4 showing the Kleiner’ score.

In addition, the following modifications were made to the manuscript:

- The phollowing sentence was added to the Legend of Fig. 2: 

(D) H&E stained liver sections observed at 20X magnification were analyzed according with semiquantitative Kleiner’s criteria.

In the Results section, the phollowing paragraph was added:

(243-249) In clinical practice, the steatosis grade (from simple steatosis to MASH) can be determined by Kleiner’s scoring system (27). According to this score, the group fed an HFS diet showed a hepatic steatohepatitis phenotype, since liver samples had ≥34% parenchymal involvement by steatosis, as well as various inflammatory foci per X20 field and several cells with a ballooning phenotype due to fat accumulation in the hepatocytes. IFC-305 administration prevented the progression of hepatic disease compared to that in the HFS group (Fig 2D).

The phollowing paragraph was added in the Discussion section:

(Line 427-433) Previous research has demonstrated an increase in the number of M1 macrophages in an experimental model of cirrhosis induced by CCl4; in that research, treatment with IFC-305 prevented this increase and favored the production of the anti-inflammatory cytokine IL-10 (48). According to these findings and the results obtained here regarding the ability of IFC-305 to prevent the increases in the levels of proinflammatory cytokines and a decrease in Kleiner’s score, which reflects lobular inflammation in metabolic syndrome, we propose that this effect is related to a regulation of immune cells.

2) The authors should measure the mRNA expression levels of lipid metabolism in the liver and Huh-7 hepatocytes. In general, amelioration of inflammation improves the dysfunction of lipid metabolism in the liver. Therefore, it is necessary to examine the mRNA expression levels of fatty acid oxidation and synthesis in the liver of the IFC-305-fed mice and the IFC-305-treated Huh-7 hepatocytes.

In the present study, we observed an important effect of the hepatoprotective IFC-305 by preventing the accumulation of lipids in the hepatocyte. Your suggestion to evaluate the mRNA of genes involved in lipid metabolism seemed very interesting. The family of enzymes responsible for the activation of fatty acids are called Long-chain acyl-CoA synthetases (ACSL). In this study we determined the expression of the ACSL1 and ACSL3 isoforms in the in vitro model of fatty acid incorporation. The results are shown in S3 Fig.

S3 Fig.

S3 Fig. mRNA expression of ACSL1 and ACSL3 in HepG2 cells. HepG2 cells were cultured in serum-free DMEM supplemented with 1% BSA- free FFA (control) in the presence of 1 mM IFC-305 or FFA with or without 1 mM IFC-305 for 24 h. Real-time quantitative polymerase chain reaction (qPCR) analysis of the mRNA expression of (A) ACSL1 and (B) ACSL3 isoforms normalized to �-actin. Relative mRNA level was calculated using the comparative ΔΔCt method. mRNA expression is expressed as mean values ± SEM of three independent experiments. **P < 0.001

The following paragraph was added to the Discussion section:

(Line 386-406) …however, this effect has since been considered to be transient, since in another study, it was found that adenosine decreases the generation of ketone bodies in a model of fatty liver induced by cycloheximide, suggesting its ability to induce β-oxidation (18). Once in hepatocytes, fatty acids that were released from adipocytes need to be activated as acyl-CoA to be metabolized, either via β-oxidation in mitochondria or by being used in TG, phospholipid or cholesterol synthesis. The family of enzymes responsible for the activation of fatty acids are called long-chain acyl-CoA synthetases (ACSLs). In this study, we determined the expression of the ACSL1 and ACSL3 isoforms in an in vitro model of fatty acid incorporation in HepG2 cells. We found that the expression of the ACSL3 isoform was decreased in cells exposed to FFAs compared to control cells, while those incubated with IFC-305 in the presence of FFAs or alone had recovery of ACSL3 expression. Regarding the ACSL1 isoform, no differences were observed, although there was a tendency toward an increase in cells incubated with IFC-305. One possibility is that the slight increases in the expression of ACSL1 and ACSL3 induced by IFC-305 favor the synthesis of acyl-CoA for oxidation and not for the synthesis of TG, since in both the in vivo and in vitro models, a decrease in the accumulation of TG deposits was observed. This is in accordance with another study in which the expression of ACSL3 was induced with oncostatin M in a model of hyperlipidemic hamsters and in HepG2 cells; in that study, greater oxidation of FFAs and a decrease in the accumulation of TG were observed (44). In another study, in hamsters fed a high-fructose diet, the expression of hepatic ACSL3 was decreased, and re-expression of ACSL3 was accompanied by a decrease in the TG level in the liver, similar to the findings with IFC-305 (45)

3) The authors should measure serum levels of the liver injury markers such as AST and ALT.

We evaluated the serum levels of AST and ALT as you recommended. There was not statistical significant difference in the results obtained, because of this,we decided to show them as a supplementary figure (S1 Fig).

The phollowing paragraph was added in the Results section

(Line 186-188) We evaluated the serum levels of AST and ALT in the different experimental groups. There was a tendency toward increased ALT in the HFS group; however, there were no statistically significant differences in the results obtained for both enzymes (S1 Fig). 

The phollowing paragraph was added in the Discussion section

(Line 336-343) It has been reported that there is no precise correlation between the serum level of the ALT enzyme and fatty liver; however, ALT has also been suggested as a possible marker (33). On the other hand, a relationship between AST and MASLD has not been reported, unlike the case for liver damage associated with alcohol consumption, in which an increase in the AST:ALT ratio is observed (34). The determination of these parameters in our metabolic syndrome model showed only a slight increase in the ALT level that was not observed in the HF+IFC-305 group, and there was no difference in the AST level in the experimental groups, suggesting a protective effect against the accumulation of fat in the liver.

4) The authors should check the minor points and English throughout the manuscript. There are many errors.

The corrected versión of the manuscript was sent to AJE.com for English style correction.

---

## [Decision Letter · Decision Letter 1]

18 Sep 2023

PONE-D-23-08771R1An adenosine derivative prevents the alterations observed in metabolic syndrome in a rat model induced by a rich high-fat diet and sucrose supplementationPLOS ONE

Dear Dr. Chagoya de Sánchez,

Thank you for submitting your manuscript to PLOS ONE. After careful consideration, we feel that it has merit but does not fully meet PLOS ONE’s publication criteria as it currently stands. Therefore, we invite you to submit a revised version of the manuscript that addresses the points raised during the review process.

We look forward to receiving your revised manuscript.

Kind regards,

Nobuyuki Takahashi, Ph.D.

Academic Editor

PLOS ONE

Journal Requirements:

Additional Editor Comments:

The revised manuscript still contains minor errors as indicated by the reviewer_1. Correct these errors before acceptance.

Reviewers' comments:

Reviewer's Responses to Questions

**Comments to the Author**

1. If the authors have adequately addressed your comments raised in a previous round of review and you feel that this manuscript is now acceptable for publication, you may indicate that here to bypass the “Comments to the Author” section, enter your conflict of interest statement in the “Confidential to Editor” section, and submit your "Accept" recommendation.

Reviewer #1: (No Response)

Reviewer #2: All comments have been addressed

2. Is the manuscript technically sound, and do the data support the conclusions?

Reviewer #1: Yes

Reviewer #2: Yes

3. Has the statistical analysis been performed appropriately and rigorously? 

Reviewer #1: Yes

Reviewer #2: I Don't Know

4. Have the authors made all data underlying the findings in their manuscript fully available?

Reviewer #1: Yes

Reviewer #2: Yes

5. Is the manuscript presented in an intelligible fashion and written in standard English?

Reviewer #1: Yes

Reviewer #2: Yes

6. Review Comments to the Author

Reviewer #1: In general, the authors revised the manuscript appropriately based on the reviewers’ comments. However, several minor mistakes are remaining.

1. Fig. 3: “INF-γ” should be revised as “IFN-γ.”

2. “N”, “L”, and “GA” are not used in Fig. 4 but explained in the legend of Fig. 4. The authors should either include them in the figure or exclude them form the legend.

Reviewer #2: The authors answered all my questions adequately.

In the future, I would like the authors to use the F4/80 analysis because it is very useful for analyzing inflammation.

7. PLOS authors have the option to publish the peer review history of their article (what does this mean?). If published, this will include your full peer review and any attached files.

Reviewer #1: No

Reviewer #2: No

---

## [Author Response · Author response to Decision Letter 1]

19 Sep 2023

Reviewer #1: In general, the authors revised the manuscript appropriately based on the reviewers’ comments. However, several minor mistakes are remaining.

1. Fig. 3: “INF-γ” should be revised as “IFN-γ.”

Answer: In Fig. 3 “INF-γ” was changed by “IFN-γ.”

2. “N”, “L”, and “GA” are not used in Fig. 4 but explained in the legend of Fig. 4. The authors should either include them in the figure or exclude them form the legend.

Answer: “N”, “L”, and “GA” were removed from the legend of Fig. 4

Reviewer #2: The authors answered all my questions adequately.

In the future, I would like the authors to use the F4/80 analysis because it is very useful for analyzing inflammation.

Thank you very much for the recommendation, we agree that the F4/80 antibody would be very useful to evaluate inflammation.

---

## [Editor Report · Decision Letter 2]

20 Sep 2023

An adenosine derivative prevents the alterations observed in metabolic syndrome in a rat model induced by a rich high-fat diet and sucrose supplementation

PONE-D-23-08771R2

Dear Dr. Chagoya de Sánchez,

We’re pleased to inform you that your manuscript has been judged scientifically suitable for publication and will be formally accepted for publication once it meets all outstanding technical requirements.

Kind regards,

Nobuyuki Takahashi, Ph.D.

Academic Editor

PLOS ONE

---

## [Editor Report · Acceptance letter]

25 Sep 2023

PONE-D-23-08771R2 

An adenosine derivative prevents the alterations observed in metabolic syndrome in a rat model induced by a rich high-fat diet and sucrose supplementation 

Dear Dr. Chagoya-de Sánchez:

I'm pleased to inform you that your manuscript has been deemed suitable for publication in PLOS ONE. Congratulations! Your manuscript is now with our production department. 

Kind regards, 

on behalf of

Dr. Nobuyuki Takahashi 

Academic Editor

PLOS ONE